# Binding pose depth modulates photoswitchable ligands' efficacy at the 5-HT₂A receptor
Verena Weber[1,2,6], Giacomo Salvadori [2,6]✉, Federico Natale [1,2], Hubert Gerwe[3], Michael Decker [3], Paolo Carloni [2]✉ & Giulia Rossetti [2,4,5]✉

Photoswitchable ligands enable reversible control of receptor signaling through light-induced *cis–trans* isomerization, yet predicting how subtle structural modifications affect efficacy remains challenging. Here, we use molecular dynamics simulations to investigate two azobenzene-based human 5–HT₂A receptor ligands differing only by a methoxy substituent position (*para–* vs *meta–*methoxy). Compound **1** (*para–*methoxy) switches from acting as a weak antagonist (*trans*) to a moderate agonist (*cis*), whereas compound **2** (*meta*-methoxy) maintains agonist activity in both forms, with *cis*-**2** exhibiting the highest efficacy. Our simulations reveal that the key determinant of these efficacy differences lies in the vertical depth of ligand insertion into the orthosteric binding pocket. The *para–*methoxy moiety of *trans–***1** forms hydrogen bonds with Asp231^5.35 and Thr160^3.37, anchoring the ligand deeper than typical tryptamine agonists and preventing engagement with activation-critical residues, thereby stabilizing the inactive receptor. Conversely, *trans–***2** lacks these anchoring interactions and adopts a shallower, agonist-compatible pose. In the active receptor, *cis–***2** forms a persistent Thr160^3.37 hydrogen bond that allows deeper penetration between TM4 and TM5, whereas *cis–***1**'s *para–*methoxy causes steric hindrances limiting this interaction. Based on these findings, we suggest that ligand insertion depth is a critical determinant of efficacy. This provides a framework for designing light-sensitive GPCR ligands with tunable signaling properties.

Light-activated drugs offer powerful tools to modulate receptor activity reversibly and selectively[1–5], enabling both the dissection of complex signaling networks and the development of therapeutics with minimal off-target effects[3,5–10]. By incorporating a photoswitchable moiety such as azobenzene into a bioactive molecule, drug activity can be reversibly controlled with specific wavelengths of light[11–15]. Upon irradiation, the azobenzene moiety undergoes *cis-trans* isomerization[16,17], inducing structural and electronic changes that can alter ligand-receptor interactions and functional outcomes at diverse targets, including G protein-coupled receptors (GPCRs)[18–25], ion channels[26], membrane transporters[27], and enzymes[28,29].

Despite such successful applications, the rational design of photoswitchable ligands exhibiting efficacy switch behavior remains challenging. Ideally, one would like to design a ligand that switches from antagonist to agonist conformation simply through light exposure. In practice, this is

done empirically[19,30–34], without knowing the molecular principles that govern whether a given photochemical isomer will act as an agonist or antagonist.

A striking illustration of this knowledge gap is provided by two recently reported photoswitchable N,N-dimethyltryptamine derivatives targeting the human serotonin 5-HT₂A receptor (Fig. 1a)[25]. The 5-HT₂A receptor is a class A GPCR with critical roles in neuropsychiatric function, serving as the primary target for both atypical antipsychotics and psychedelic compounds[35,36]. These two closely related compounds, *para*-methoxy-5-azo-DMT (**1**, Fig. 1b) and *meta*-methoxy-5-azo-DMT (**2**, Fig. 1b), differ only in the position of a single methoxy substituent on the azobenzene moiety. Yet, this seemingly minor structural variation produces dramatically different pharmacological outcomes upon photoisomerization. In cell-based β-arrestin recruitment assays[25], **1** behaves almost like an on/off switch:

[1]RWTH Aachen University, Aachen, Germany. [2]Computational Biomedicine, Institute for Neuroscience and Medicine INM-9, Forschungszentrum Jülich GmbH, Jülich, Germany. [3]Julius-Maximilians-Universität Würzburg (JMU), Institut für Pharmazie und Lebensmittelchemie, Pharmazeutische und Medizinische Chemie, Würzburg, Germany. [4]Department of Neurology, Medical Faculty, RWTH Aachen University, Aachen, Germany. [5]Jülich Supercomputing Center (JSC), Forschungszentrum Jülich, Jülich, Germany. [6]These authors contributed equally: Verena Weber, Giacomo Salvadori. ✉e-mail: g.salvadori@fz-juelich.de; p.carloni@fz-juelich.de; g.rossetti@fz-juelich.de

**Fig. 1 | The 5-HT$_{2A}$ receptor system and the photoswitchable ligands. a** Representation of the X-ray inactive (PDB id: 6A94[44]) and cryo-EM G$_q$-coupled active (PDB id: 6WHA[45]) 5-HT$_{2A}$ structures. **b** Chemical structures of the photoswitchable ligands *para*-methoxy-5-azoDMT (**1**) and *meta*-methoxy-5-azoDMT (**2**) in their *trans* configuration, with atom numbering used throughout. The labels of the ligand atoms used in the manuscript are also reported. **c** Key microswitches analyzed in this work, including the Trp336[6.48] toggle switch, Arg173[3.50]-Glu318[6.30] ionic lock, Cys322[6.34]-Arg173[3.50] TM6 displacement, and Na$^+$ pocket.

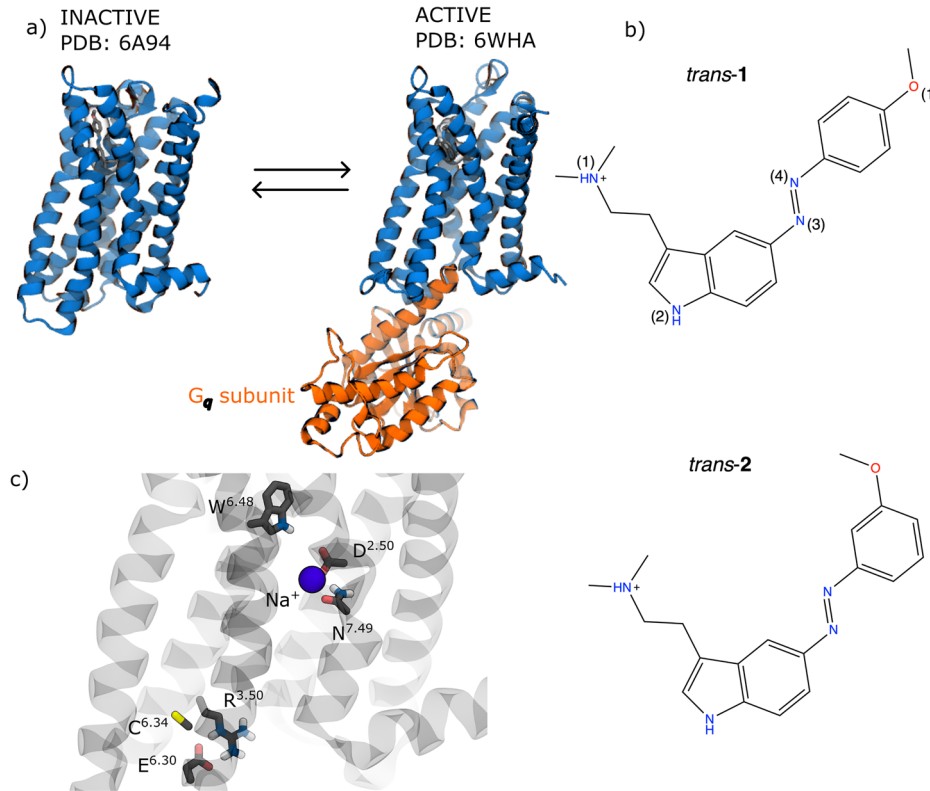

its *cis* isomer exhibits moderate agonist activity (~45% of the response elicited by the partial agonist LSD), whereas its *trans* isomer yields only weak antagonism (~20% of LSD's maximal response). In contrast, **2** retains substantial agonist activity in both isomeric states (*cis* ~65% and *trans* ~55% of LSD's response), failing to achieve the desired "off" state. This inability to fully suppress receptor activation severely limits its pharmacological applications.

Recent evidence suggests that the exact binding mode of a ligand within the orthosteric binding pocket of a GPCR can dramatically influence its signaling outcome[37]. Structural studies have shown that traditional, non-photoactive ligands may engage either deeply buried subpockets or more superficial vestibular regions of the receptor, and that this binding pose diversity can modulate ligand affinity, efficacy, and even establish a signaling bias in GPCRs other than 5-HT$_{2A}$[38–40]. Recently, a photoswitchable drug was designed to achieve signaling bias in the 5-HT$_{2A}$ receptor itself[41]. Based on these insights, we hypothesized that the divergent efficacy profiles of compounds **1** and **2** towards 5–HT$_{2A}$ arise also from isomer-specific differences in the binding depth of the receptor pocket.

To test this hypothesis, we performed all-atom molecular dynamics (MD) simulations of compounds **1** and **2** in both *cis* and *trans* configurations (*trans*–**1**, *cis*–**1**, *trans*–**2**, and *cis*–**2**) bound to 5-HT$_{2A}$ inactive and active states. Our simulations reveal that the *para* versus *meta* positioning of the methoxy substituent fundamentally alters ligand anchoring within the binding pocket. In the inactive receptor, the *para*-methoxy group of *trans*-**1** enables formation of a hydrogen bond network involving Asp231[5.35] in extracellular loop 2 and Thr160[3.37] at the base of the orthosteric pocket (Fig. 2). This dual anchoring mechanism pulls *trans*-**1** significantly deeper into the binding site compared to *trans*-**2**, which lacks these stabilizing interactions due to its *meta* substitution pattern. After photoisomerization, both *cis* isomers adopt shallower, agonist-like poses that destabilize the inactive state; a similar behavior can also be found for the *trans* isomers in the active receptor. Instead, *cis*-**2** forms a persistent hydrogen bond with Thr160[3.37] in the active receptor that allows for a deeper penetration in between TM4 and TM5, whereas *cis*-**1** cannot form this interaction.

Together, our findings reveal the binding pose depth as a key determinant of efficacy for photoswitchable 5-HT$_{2A}$ ligands, providing a mechanistic foundation for the rational design of photopharmacological switches for this receptor and potentially other GPCRs with similar structural motifs to those of 5-HT$_{2A}$[31,42,43].

## Results

### Simulations of ligand-receptor complexes

To understand why compounds **1** and **2** exhibit such different pharmacological profiles despite their structural similarity, we performed eight independent MD simulations of the 5-HT$_{2A}$ receptor: each photoswitchable ligand (**1** or **2**) in each isomeric form (*trans* or *cis*) was simulated in both the inactive and active receptor states. This approach yielded eight distinct systems with a cumulative simulation time of ~80 microseconds (Table S1). The inactive receptor model was based on the antagonist risperidone-bound X-ray structure (PDB ID: 6A94[44]), while the active state model was derived from the G$_q$-coupled 5-HT$_{2A}$ cryo-EM structure (PDB ID: 6WHA[45]). Importantly, we retained the G$_q$ subunit in all active-state simulations, as preliminary MDs showed that its absence caused the receptor to relax towards an inactive-like conformation[46]. Each receptor complex was embedded in a neuronal-mimetic lipid bilayer, and initial ligand-receptor poses were generated by constrained molecular docking (see Methods and Supplementary Note 1 for detailed protocols). In all cases, the docked pose preserved the hallmark salt bridge between the ligand's protonated amine and Asp155[3.32], a conserved anchor for aminergic GPCR ligands.

### Activation microswitches

GPCRs transition between inactive and active states through coordinated conformational changes in conserved "microswitches", structural motifs that relay ligand binding to G protein coupling. Activation involves several key events: rotation of the Trp336[6.48] toggle switch, breaking of the Arg173[3.50]–Glu318[6.30] ionic lock, displacement of an allosteric Na$^+$ ion, and the outward movement of transmembrane helix 6 (TM6) that enables G protein or β-arrestin binding[47–55].

**Fig. 2 | Hydrogen bond network analysis in the inactive receptor.** Kernel density estimation (KDE) distribution of the H-bonding interactions with Thr160[3.37], Ser242[5.46], Asp231[5.35] (O1–H-N-Asp), and Asn[6.55] residues for (**a**) *trans*-**1**, (**b**) *trans*-**2**, (**c**) *cis*-**1**, and (**d**) *cis*-**2** in the inactive receptor. Insets show representative binding poses. Analysis was performed on trajectory frames extracted every 1 ns.

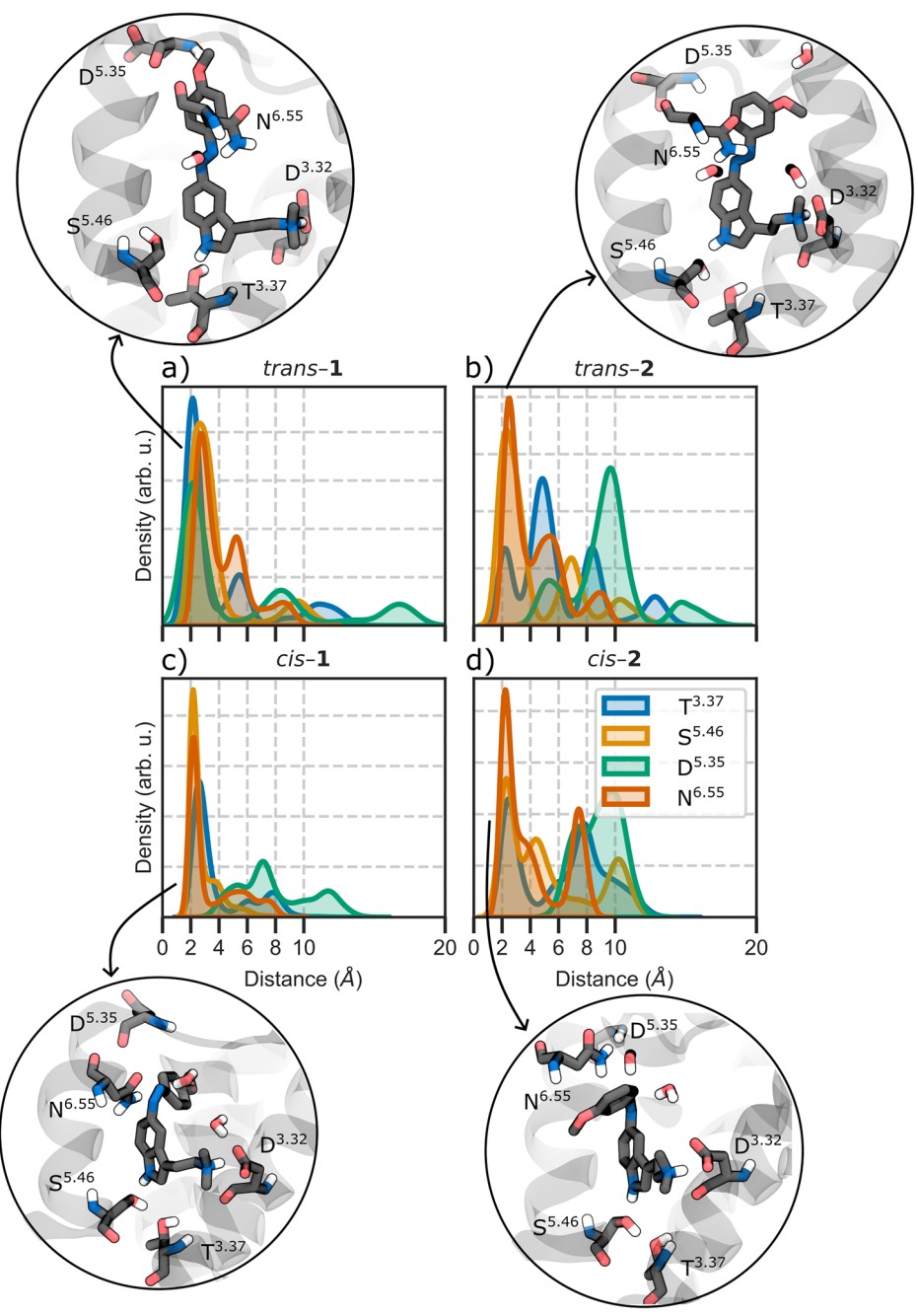

We therefore monitored these activation markers: (i) the $\chi_1$ and $\chi_2$ dihedral angles of Trp336[6.48], (ii) the local environment of the sodium pocket, (iii) the Arg[3.50]–Glu[6.30] ionic lock ($d_{IL}$), and (iv) the TM3–TM6 cytoplasmic gap ($d_{TM}$). These metrics distinguish inactive states ($d_{TM} \approx 9$ Å, $d_{IL} \approx 4$ Å, $\chi_1 \approx -80°$, Na[+] coordinated) from active states ($d_{TM} \approx 12$ Å, $d_{IL} \approx 16$ Å, $\chi_1 \approx -160°$, Na[+] expelled).

To determine whether our simulated binding poses were compatible with receptor activation, we compared them to experimentally determined agonist binding modes. We used the tetracyclic core of LSD, a well–characterized partial agonist, as our reference, aligning against structures in (i) the fully active cryo-EM structure (PDB 9AS3[56]) and (ii) the X-ray crystal structure in the absence of G-protein that resembles features of the inactive state (PDB 6WGT[45]). By calculating the root-mean-square deviation of each ligand's indole core relative to LSD (RMSD$_i$), we could quantify how closely our compounds mimicked established agonist binding. Poses maintaining RMSD$_i$ < 2 Å were considered to adopt similar,

potentially activation-compatible orientations. Here, we discuss only the long-lived binding modes; transient poses are detailed in Supplementary Note 2.

**Trans isomers exhibit opposite effects in the inactive receptor through distinct binding depths**

In the inactive receptor, *trans*-**1** establishes a unique hydrogen bond network involving its *para*-methoxy group and the backbone carbonyl of Asp231[5.35] (Fig. 2, S1, 43% of occurrence overall the MD trajectories), combined with the indole $N_1$-H bond to Thr160[3.37] (70% occupancy) and the conserved salt bridge to Asp155[3.32] (99% occupancy). This triad of interactions creates a rigid binding pose with minimal fluctuation (Fig. S2). Extensive hydrophobic contacts (Val235[5.39], Val156[3.33], Ser159[3.36], and Leu230 are present in ≥ 75% of frames) further stabilize this pose. This interaction network (Asp231[5.35] and Thr160[3.37]) pulls *trans*-**1** deeper into the orthosteric pocket compared to typical tryptamine agonists such as LSD,

**Fig. 3 | *Cis*-2 triggers toggle switch rotation in the inactive receptor.** Moving average (window of 20 frames) of the time evolution of four intermolecular distances. Key aromatic or polar side chains are shown as sticks and labeled by the Ballesteros-Weinstein index. Top panel, three representative snapshots extracted from the distance plot. **a** Initial state: the indole N1–H of *cis*-2 hydrogen-bonds to either Thr160$^{3.37}$ or Ser242$^{5.46}$ while Phe$^{5.47}$ stacks with Phe340$^{6.52}$, maintaining Trp$^{6.48}$ in its vertical, inactive rotamer. **b** The ligand interacts with Trp$^{6.48}$, breaking the Phe$^{5.47}$–Phe340$^{6.52}$ π-stack. **c** Trp$^{6.48}$ completes its flip and establishes a π–π interaction with Phe$^{5.47}$, while *cis*-2 anchors to Ser$^{5.43}$. The behavior of the toggle switch in the other systems is reported in Fig. S5. Analysis was performed on trajectory frames extracted every 1 ns.

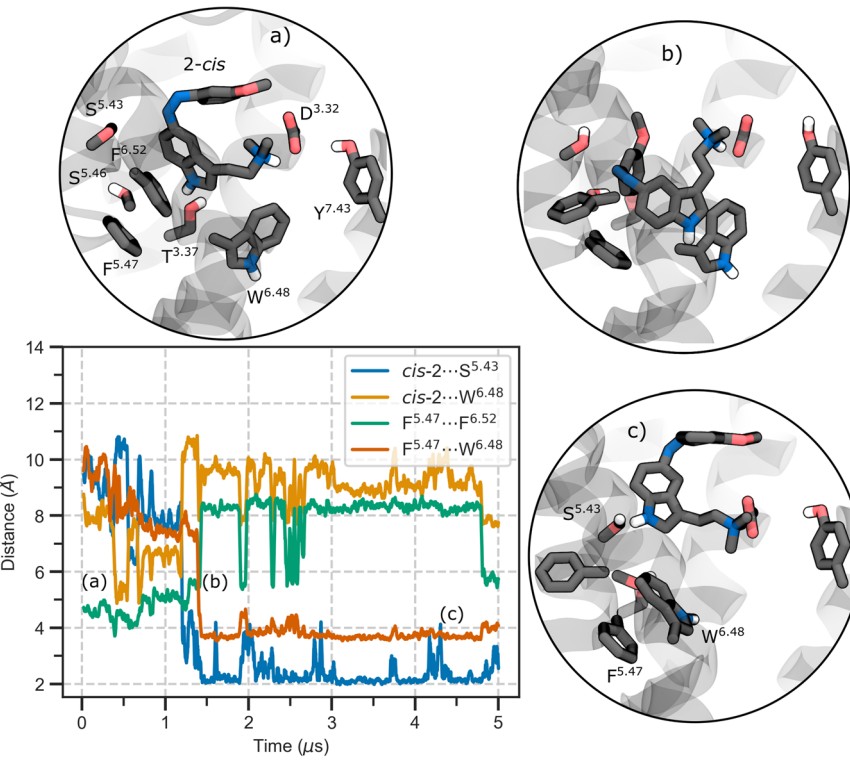

DMT, and psilocin[56]. This over-insertion prevents *trans*-1 from engaging the polar triad, i.e., Ser239$^{5.44}$, Ser242$^{5.46}$, and Asn343$^{6.55}$, known to be critical for agonist activity at the 5-HT$_{2A}$ receptor[57]. The latter, in particular, has been shown to be essential for 5-HT's agonist potency[45]. The incompatibility of this orientation with receptor activation is also supported by multiple metrics: RMSD$_i$ > 2.0 Å for 55% of the simulation time, narrow TM3-TM6 distance, intact ionic lock, and vertical toggle switch orientation (Fig. S3–S5, Table S2). Together, these observations indicate that *trans*-1 stabilizes the receptor's inactive conformation.

In contrast, *trans*-2 cannot form the same anchoring interactions due to its *meta*-methoxy position and consequently adopts a different binding mode (Fig. 2b). *Trans*-2 maintains the Asp155$^{3.32}$ salt bridge, but exhibits greater conformational flexibility within the pocket and adopts a pose closely resembling that of the partial agonist LSD in the active receptor (RMSD$_i$ <2 Å for 78% of simulation time). This allows intermittent engagement of the canonical polar network (Fig. S1): *trans*-2 transiently forms H-bonds with Thr160$^{3.37}$ (~10% occupancy) and Ser242$^{5.46}$ (~20%), while still maintaining hydrophobic contacts with Leu229, Leu230, Val235$^{5.39}$, Val156$^{3.33}$, and Val366$^{7.40}$ (each ~50–65% occupancy). Overall, however, *trans*-2 on its own did not trigger substantial receptor activation on the simulation timescale, keeping a vertical toggle switch, narrow TM3-TM6, and closed ionic lock (see Fig. S3–S5, Table S2).

**Cis isomers partially destabilize the inactive receptor**

The *cis* isomers, upon photoisomerization from *trans*, adopt shallower binding poses, similar to those adopted by LSD. *Cis*-1 maintains the critical Asp155$^{3.32}$ salt bridge (as expected for tryptamine analogs), but it is no longer able to reach Asp231$^{5.35}$, and its binding pose shifted upward relative to *trans*-1. Consequently, *cis*-1 partially restored some agonist-like contacts: it forms stable hydrogen bonds with Ser239$^{5.44}$ and Ser242$^{5.46}$ (each ~50–70% occupancy) and even engaged Asn343$^{6.55}$ (Fig. 2c, S1). Indeed, the indole ring of *cis*-1 overlays with the tetracyclic core of LSD for ~75% of the trajectory frames (RMSD$_i$ <2 Å), highlighting that *cis*-1 binds in an LSD-like orientation in the inactive state. *Cis*-1 also engages a network of hydrophobic contacts with residues such as Phe234, Gly238$^{5.43}$, Trp336$^{6.48}$, Val156$^{3.33}$, and Leu229 (occurrences ~45–87%). As a result, this gives *cis*-1 a

well-anchored yet not over-deep pose. However, *cis*-1 does not fully activate the receptor on its own (see Fig. S4–S5, Table S2). The Trp336$^{6.48}$ toggle also stays in the inactive (vertical) rotamer in almost all trajectories. Only in one case, the temporary cleavage of the *cis*-1-Ser239$^{5.44}$ hydrogen bond causes the ligand to reposition within the binding pocket. This repositioning led to a temporary loss of π–π stacking between Phe243$^{5.47}$ and Phe340$^{6.52}$, two aromatic residues that line the binding pocket. Consequently, Trp336$^{6.48}$ was able to flip to its horizontal, active-like rotamer (Fig. S5, S22).

*Cis*-2, on the other hand, has the most pronounced agonist-like effect on the inactive receptor. In its inactive-state simulations, *cis*-2 maintains a similar hydrophobic contact pattern to *cis*-1 (e.g., with Val156$^{3.33}$, Leu229, Trp336$^{6.48}$, occurrences ~50–87%). These contacts keep *cis*-2 embedded in the orthosteric site, but interestingly, *cis*-2 exhibits greater mobility within the pocket than *cis*-1 (Fig. S2). This flexibility correlates with weaker polar interactions: *cis*-2's hydrogen bonds to Ser242$^{5.46}$ and Asn343$^{6.55}$ are much less frequent (only ~20% and ~28% occupancy, respectively, vs. >50% for *cis*-1, see Fig. 2d, S1), and even the Asp155$^{3.32}$ salt-bridge occupancy is slightly reduced (~80% for *cis*-2 vs. ~99% for *cis*-1, Fig. S1). The increased mobility of *cis*-2 induces a rotation of the Trp336$^{6.48}$ toggle switch from the vertical (inactive) to the "horizontal" (active) rotameric states (Fig. 3, S5). In fact, in more than 75% of simulation frames, the indole core of *cis*-2 aligns closely with LSD's ergoline core, indicating an agonist-like pose. Upon the complete loss of Ser242$^{5.46}$ and Asn343$^{6.55}$ H-bonds, *cis*-2 experiences increased mobility and a slight drift. This allows it to re-anchor to Ser239$^{5.44}$, simultaneously prompting Trp336$^{6.48}$ to swing downward into an active-like orientation (Fig. 3). This mechanistic shift is facilitated by the disruption of π–π stacking between Phe340$^{6.52}$ and Phe243$^{5.47}$, creating space for Trp336$^{6.48}$ to rotate downward and establish a π–π interaction with Phe243$^{5.47}$.

However, even though *cis*-2 clearly destabilizes the inactive receptor by weakening the ionic lock (Fig. S3, Table S2), we did not observe a full TM6 outward movement (Fig. S4, Table S2). In other words, *cis*-2 on its own presents the strongest agonist effect in the inactive receptor (by flipping Trp336$^{6.48}$), but complete activation still likely requires the presence of the G protein or other factors.

While the binding modes described above represent the dominant conformations observed across replicas, our simulations also revealed

**Fig. 4 | Hydration of the allosteric sodium pocket in the inactive receptor. a** The continuous water pathway is blocked, leaving the Na$^+$ ion (blue sphere) tightly coordinated by two water molecules, Asp$^{2.50}$ and Asn$^{7.49}$, and Phe$^{6.44}$ sidechains.
**b** Disruption of the Asn$^{7.49}$–Asp$^{2.50}$ hydrogen bond and an outward shift of Phe$^{6.44}$ open a solvent pathway from the extracellular vestibule to the sodium pocket. Gray cartoons depict the seven-TM bundle; insets enlarge the Na$^+$ site with key residues labeled.

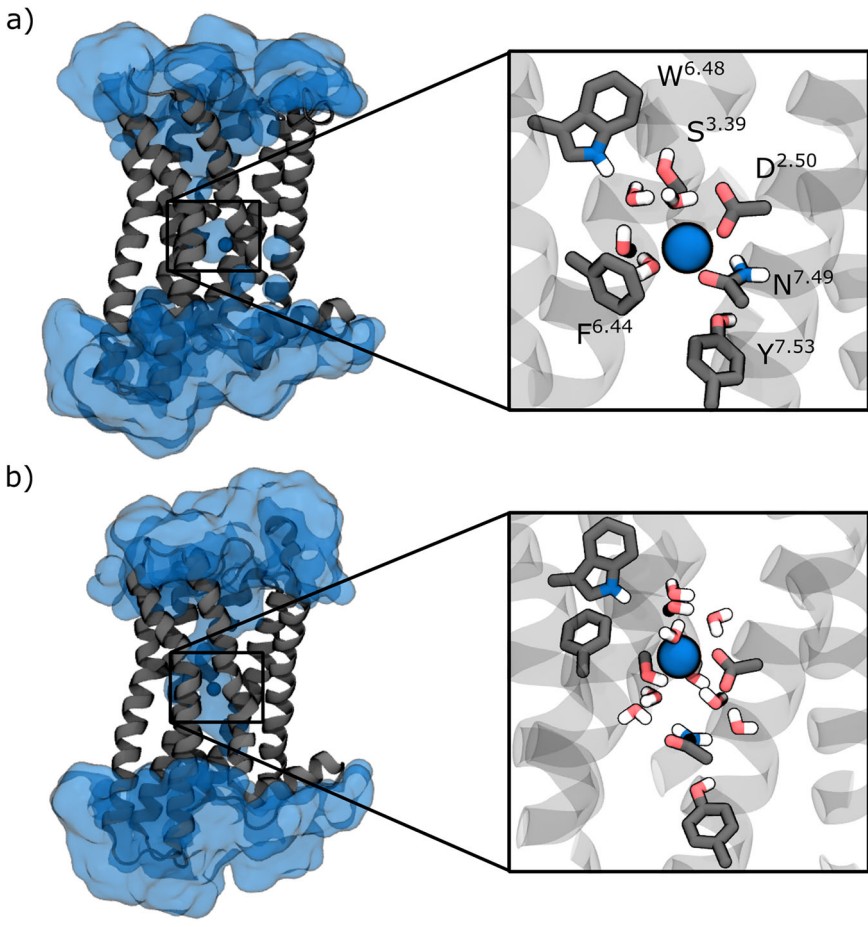

replica-specific secondary poses detailed in Supplementary Note 2. For *trans*-**1** in the inactive receptor, transient disruption of the Asp231$^{5.35}$ interaction allowed the ligand to tilt toward TM5, forming brief contacts with Ser239$^{5.44}$ and Gly238$^{5.43}$. One replica of *trans*-**2** sampled a vestibular configuration characterized by an upward shift within the binding pocket and weakening of the Asp155$^{3.32}$ salt bridge. For *cis*-**1**, rotation around the diazo bond in one replica disrupted the Ser242$^{5.46}$ and Asn343$^{6.55}$ hydrogen bonds, inducing a reorientation of the indole ring toward Ser239$^{5.44}$ or Gly238$^{5.43}$. *Cis*-**2** showed the largest conformational variability: in two replicas, the ligand transiently disengaged from the conserved Asp155$^{3.32}$ anchor, drifting into the extracellular vestibule and adopting a weakly bound, solvent-stabilized pose. These reflect the inherent conformational flexibility of the receptor-ligand systems, leading to alternative configurations that reflect local rearrangements rather than substantive divergence between simulations.

### Different effects on the allosteric sodium pocket on the inactive receptor

Differences in ligand binding were also reflected in the allosteric sodium (Na$^+$) site and local water distribution, which are known to correlate with GPCR activation[58]. In all our inactive-state simulations, a Na$^+$ ion remained bound to the side chain of the conserved residue Asp120$^{2.50}$, located deep in the receptor core. However, the coordination and hydration of this Na$^+$ pocket varied in a ligand-dependent manner (Fig. S6–S9).

With *trans*-**1** bound, Na$^+$ is trapped within a hydrophobic cage formed by Phe332$^{6.44}$, which acts as a lid above the ion, while Asp120$^{2.50}$ provides direct electrostatic anchoring. This coordination is further stabilized by Asn380$^{7.49}$, which forms a critical hydrogen bond with Asp120$^{2.50}$ that locks the sodium-binding residue in an optimal geometry (Fig. 4a). When this stabilizing network remains intact (approximately two-thirds of the

simulation time, see Fig. S6), the sodium ion maintains a restricted coordination sphere with only two water molecules on average, creating a relatively dehydrated pocket characteristic of the inactive state. However, either the disruption of the Asn380$^{7.49}$-Asp120$^{2.50}$ hydrogen bond or an outward shift of Phe$^{6.44}$ triggers a cascade of structural changes: the loss of this interaction destabilizes the Asp120$^{2.50}$ side chain orientation, weakening its grip on Na$^+$, while simultaneously breaking the Na$^+$-Asn380$^{7.49}$ and Na$^+$-Phe332$^{6.44}$ coordination bonds. This conformational change opens a continuous water channel from the extracellular vestibule to the sodium pocket, dramatically increasing the Na$^+$ hydration shell to four water molecules on average (Fig. 4b). For *trans*-**2**, this hydration event occurs almost immediately, within the first microsecond of dynamics (Fig. S7), reflecting the ligand's inability to stabilize the inactive sodium pocket geometry. In essence, *trans*-**2** binding primes the sodium microswitch for activation: the ion remains bound to Asp120$^{2.50}$, but the pocket is looser and populated by a higher number of water molecules, a state linked to enhanced receptor readiness to transition toward activation. *Trans*-**1**, in contrast, resists this priming: it keeps the Na$^+$ pocket "closed" and dehydrated far longer (Fig. S6).

*Cis*-**1** shows a slight increase in water accessibility around its binding site compared to *trans*-**1**. On average, ~5 water molecules are present in the immediate vicinity of *cis*-**1** (mostly around the azobenzene linkage), versus essentially zero water molecules for *trans*-**1**. Notably, even during these momentary perturbations, *cis*-**1** maintains a grip on the sodium site: like *trans*-**1**, it preserves the Asn380$^{7.49}$–Asp120$^{2.50}$ interaction and keeps Na$^+$ coordinated by only ~2 water molecules. Furthermore, *cis*-**1** exhibits a unique "backup" Na$^+$ coordination via Ser162$^{3.39}$, which transiently steps in to ligate Na$^+$ whenever the Asp120$^{2.50}$-Asn380$^{7.49}$-Na$^+$ network transiently disrupts. This secondary sodium coordination prevented full hydration of the pocket and in fact results in even lower average Na$^+$ hydration for *cis*-**1**

**Fig. 5 | Hydrogen bond network analysis in the active receptor.** KDE distribution of the H-bonding interactions with Thr160$^{3.37}$, Ser242$^{5.46}$, Asp231$^{5.35}$, and Asn343$^{6.55}$ residues, together with a close-by representation of (**a**) *trans*-**1**, (**b**) *trans*-**2**, (**c**) *cis*-**1**, and (**d**) *cis*-**2** with their main H-bonding interactions in the active receptor. Analysis was performed on trajectory frames extracted every 1 ns.

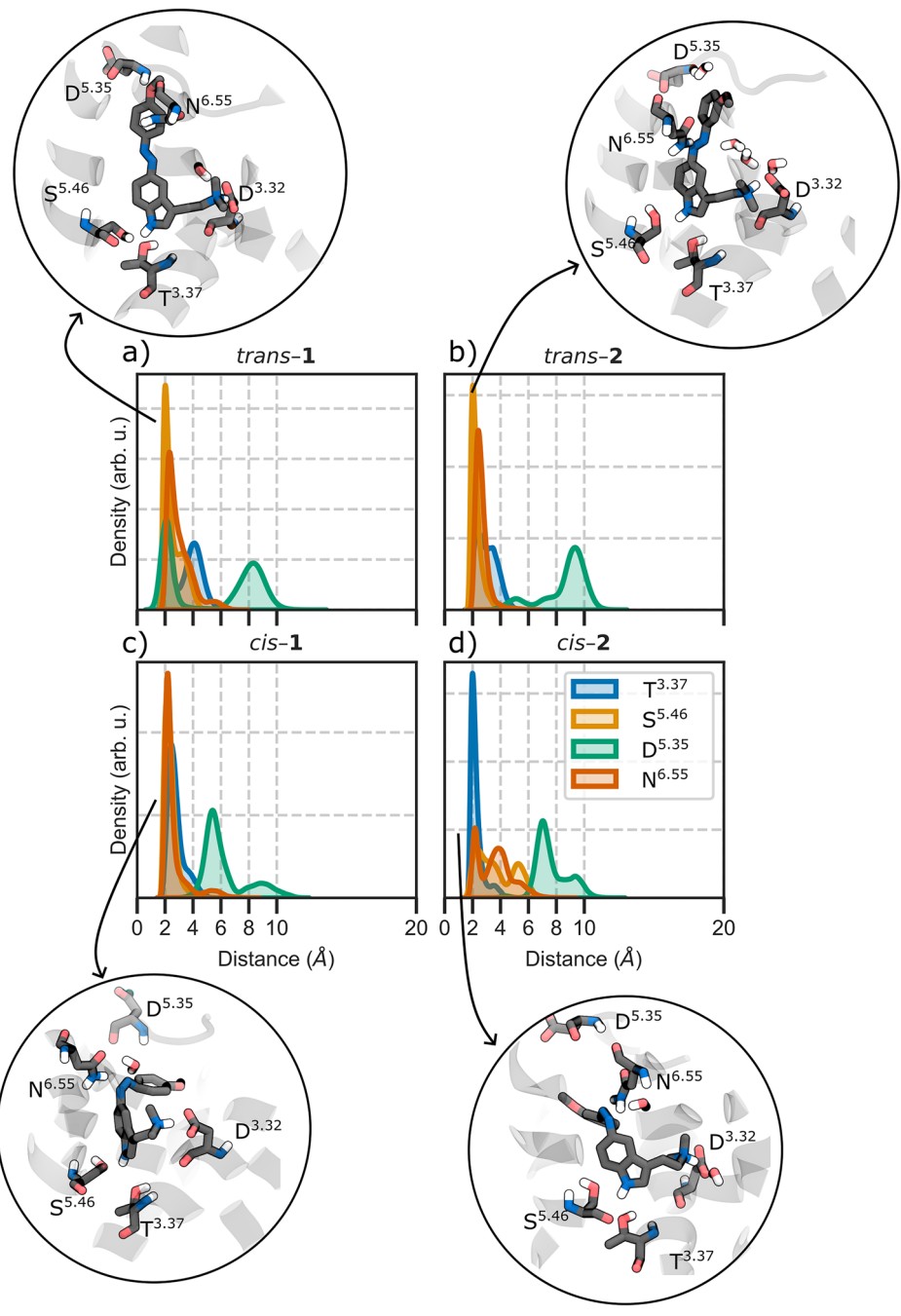

than for *trans*-**1**. Only the cleavage of both the primary (Asp–Asn) and secondary (Ser162$^{3.39}$) Na$^+$ coordination allows a rush of water into the pocket (Fig. S8).

In *cis*-**2**, when the Asn380$^{7.49}$–Asp120$^{2.50}$–Ser162$^{3.39}$ triad is intact, and Phe340$^{6.52}$ adopts its closed rotamer, Na$^+$ remains tightly bound in a two-water coordination shell (Fig. S9). A first hydration event occurs when Trp336$^{6.48}$ flips to its horizontal rotamer, creating a solvent path that admits one or two extra water molecules while the Asn380$^{7.49}$–Asp120$^{2.50}$ interaction is still strong (Fig. S9, middle panel). The most pronounced hydration, however, is seen when both the Asn380$^{7.49}$ contact loosens, and Phe340$^{6.52}$ flips to its open rotamer: the aromatic lid lifts, the conduit widens, and a continuous water channel from the extracellular vestibule to Asp120$^{2.50}$ forms. In these frames, the Na$^+$ hydration shell expands to 4–5 waters (Fig. 4b, S9).

## Ligands binding in the active-state receptor

We next examined the behavior of each ligand in the active 5-HT$_{2A}$ receptor. All four ligand isomers (*cis*-**1**, *trans*-**1**, *cis*-**2**, and *trans*-**2**) consistently stabilized the active 5-HT$_{2A}$ state, as can be seen from $d_{TM}$ and $d_{IL}$ (Fig. S10, S11, Table S2).

In the active complex, *trans*-**1** toggles between its Asp231-anchored pose (39%, weakened because the receptor's extracellular vestibule has widened) and an LSD-like orientation that restores the Ser242$^{5.46}$ and Asn343$^{6.55}$ H-bonds (70% and 34%, respectively, Fig. 5a, S12). Its salt bridge persists for 99% of the simulation. The ligand establishes hydrophobic contacts with Leu228 (51%), Val235$^{5.39}$ (76%), Val156$^{3.33}$ (74%), Ser159$^{3.36}$ (93%), and Trp336$^{6.48}$ (52%), contributing to the overall stability of the binding pose within the active receptor conformation. A RMSD$_i$ < 2 Å was observed for 85% of the simulation time.

Within the active receptor conformation, *trans*-**2** forms an H-bond with the sidechain of Ser242$^{5.46}$ (70% occurrence), a key interaction for many 5-HT$_{2A}$ agonists (Fig. 5b)[57]. The ligand's orientation within the orthosteric pocket is reinforced by sustained contacts with Val235$^{5.39}$ (75%), Ser159$^{3.36}$ (93%), Ser242$^{5.46}$ (95%), and Tyr370$^{7.42}$ (76%). Similar to *trans*-**1**, also the binding pose of *trans*-**2** adopted in the active receptor state exhibited RMSD$_i$ < 2 Å in 88% of simulated time.

The *cis*-**1** isomer bound to the active receptor exhibits a hydrogen bond with Ser239$^{5.44}$ (Figs. 5c, S12, 54% of occurrence) and maintains frequent interactions with Asn343$^{6.55}$ (49% of occurrence). As for the *trans* isomer, *cis*-**1** establishes a salt bridge between the tertiary amine and Asp155$^{3.32}$ (99%). This binding pose is further stabilized by persistent hydrophobic interactions with Ser159$^{3.36}$ (87%), Val156$^{3.33}$ (87%), Phe234 (66%), and Phe340$^{6.52}$ (82.5%), reinforcing its orientation within the orthosteric pocket. In fact, the positioning of the indole moiety *cis*-**1** closely resembles that of LSD in the active receptor state, with RMSD$_i$ values below 2.0 Å in over 92% of simulation frames. *Cis*-**2**, by contrast, penetrates the binding pocket in the active receptor. The *meta*-methoxy substituent of *cis*-**2** reaches deeper into the cavity between TM4 and TM5 (where *cis*-**1**'s *para* substituent could not), and as a result *cis*-**2** forms a noteworthy hydrogen bond with Thr160$^{3.37}$ (on TM3) ~ 66% of the time in the active state. *Cis*-**2**'s interaction with Thr160$^{3.37}$, a residue near the base of the orthosteric site, appeared to compensate for its relatively weak hydrogen bond with Ser242$^{5.46}$ in the active state (only ~16% occupancy for *cis*-**2** vs. ~62% occupancy for *cis*-**1**).

Overall, the active-state simulations reinforce that all four isomers are capable of accommodating and stabilizing the active 5-HT$_{2A}$ receptor. None of the ligands acts as an inverse agonist or destabilizer of the active state; rather, indicating that the efficacy differences manifest primarily in the inactive-state interactions described above. In other words, when the receptor is forced into the active state (by G protein or other factors), even *trans*-**1** can bind in a way that does not prevent activation, but when the receptor is inactive, only certain ligand poses can trigger its activation.

### Toggle switch behavior in the active receptor

Our simulations offer valuable insights into the behavior of the toggle switch residue Trp336$^{6.48}$, whose reorientation is considered a key step in Class A GPCR activation. Notably, during the equilibration phase of our simulations of the active receptor, Trp336$^{6.48}$ transitions from its initial horizontal orientation to a vertical conformation characteristic of the inactive receptor state in the presence of all our ligands (Fig. S13), yet the receptor was clearly active by all other measures (Fig. S10, S11). Likewise, recent high-resolution structures[56] show a similar vertical orientation in the active 5-HT$_{2A}$ structure bound to either LSD or serotonin (Fig. S14a, b)[56], even though the receptor is fully active.

Rotation of Trp336$^{6.48}$, though structurally feasible with **1** and **2**, does not occur during all our simulations of the active receptor. This is in line with what observed for both ergoline and tryptamine chemotypes, which share a vertical orientation and lack the bulky N-benzyl or phenoxy extensions that would displace the toggle switch (Fig. S14c). Conversely, when *cis*-**2** is placed in the inactive receptor, we occasionally observed Trp336$^{6.48}$ adopting a downward rotamer (Fig. 3) reminiscent of the orientation captured in structures of bulkier, β-arrestin-biased agonists such as 25-CN-NBOH and RS130-180, whose extended N-benzyl arms press directly on Trp336$^{6.48}$ (Fig. S14c).

However, we must acknowledge that our microsecond-scale simulations may not fully capture the complete energetic landscape of toggle switch transitions. The free energy barriers separating different Trp336$^{6.48}$ rotameric states and the precise transition pathways between them would require enhanced sampling methods to characterize comprehensively. Furthermore, while crystal structures provide invaluable snapshots of stable conformational states, they necessarily capture only single, low-energy conformations within what may be a broader ensemble of active states. The crystallization conditions, temperature, and crystal packing forces may also influence which conformations are observed. Our classical molecular

dynamics approach, while extensive, operates within the constraints of predetermined force field parameters that may not fully account for the electronic polarization effects or subtle π-π interactions that could influence toggle switch stability. These limitations suggest that the vertical toggle switch orientation we observe may represent one of several possible active state conformations, with the full conformational heterogeneity and dynamic equilibria between states remaining to be fully elucidated through complementary experimental and computational approaches.

### Discussion

Functional assays had shown that **1** (*para*-methoxy) behaves almost like an on/off switch[25]: its *cis* isomer activates the receptor moderately (partial agonist), whereas its *trans* isomer elicits minimal activity (antagonist-like). In contrast, **2** (*meta*-methoxy) retains substantial agonist activity in both *cis* and *trans* forms. Our simulations underscore that the vertical depth of ligand insertion within the orthosteric pocket is a critical determinant for 5-HT$_{2A}$ activation efficacy, acting in concert with specific residue interactions.

In the inactive receptor, *trans*-**1**'s *para*-methoxy group enables formation of a unique hydrogen bond with Asp231$^{5.35}$ that, combined with Thr160$^{3.37}$ anchoring, facilitates a deeper insertion of the ligand into the binding pocket than those of typical agonists. This over-insertion positions *trans*-**1** below Ser239$^{5.44}$ and Ser242$^{5.46}$ required for activation, effectively locking the receptor in an inactive conformation (Fig. 6b).

*Trans*-**2**'s meta-methoxy substitution prevents this deep anchoring, allowing it to maintain an LSD-like pose at the appropriate depth for engaging activation-critical residues (Fig. 6a). While both *trans* isomers can bind the active receptor in agonist-compatible poses, only *trans*-**1**'s ability to adopt the deeply anchored pose in the inactive state explains its antagonist-like behavior in functional assays. The vertical depth of ligand insertion thus emerges as a key determinant of the efficacy among the two *trans* conformers.

*Cis*-**1**, when bound to the active receptor, loses the Asp231$^{5.35}$ anchor and adopts a more elevated pose within the orthosteric pocket compared to its deeply-bound *trans*-**1** counterpart in the inactive state (Fig. 6b): *cis*-**1**'s *para*-methoxy group and the distal phenyl ring of its azobenzene moiety encounter steric limitations for deep entry into the hydrophobic tunnel between TM4 and TM5[59] (Fig. S15). This contrasts to *cis*-**2**, whose *meta*-methoxy group penetrates deeply into the hydrophobic tunnel of the active receptor (Fig. 6a, S15), leading to the formation of an H-bond with Thr160$^{3.37}$, and a deeper insertion in the binding site. Such observed deeper binding was also found for some high-affinity 5-HT$_{2A}$ agonists, which do bind in this region[44]. Such evidence suggests that the higher agonistic power of *cis*-**2** with respect to *cis*-**1** is linked to the depth of the binding pose. Like for the *trans* conformers, also for the *cis* ones, the vertical depth of ligand insertion emerges as a key determinant of the efficacy.

Our findings align with and extend recent structural insights showing that subtle variations in GPCR binding poses can produce dramatically different signaling outcomes[37,60].

This insight aligns with previous structural observations showing that the specific binding depth resembling the one of LSD is associated with agonism, whereas deviations are linked to receptor inactivation and antagonism[44,60]. Notably, deep insertion can stabilize either active or inactive conformations depending on the receptor state, highlighting the importance of context-specific ligand positioning for functional outcomes. The observation that antagonists like risperidone and zotepine bind even deeper than *trans*-**1** (Fig. S16) supports the notion that excessive insertion depth can hinder activation-relevant contacts and favor receptor inactivation[44]. This hypothesis is further supported by mutational data on the related 5-HT$_{2B}$ receptor[60], where increasing cavity space enabled antagonist-to-agonist conversion by promoting an LSD-like binding pose. Mutational simulations of D155A, T160A, and D231P further corroborate the cooperative nature of this anchoring network: removal of individual residues destabilizes

**Fig. 6 | Schematic overview of depth-dependent efficacy switching. a** In the inactive receptor (blue), *trans*-**2** (cyan) maintains an agonist-like shallow pose while *cis*-**2** (purple), in the active receptor (orange), achieves the deepest insertion via Thr160[3.37] anchoring. **b** *Trans*-**1** (cyan) in the inactive receptor inserts deeply, while *cis*-**2** maintains a shallower pose.

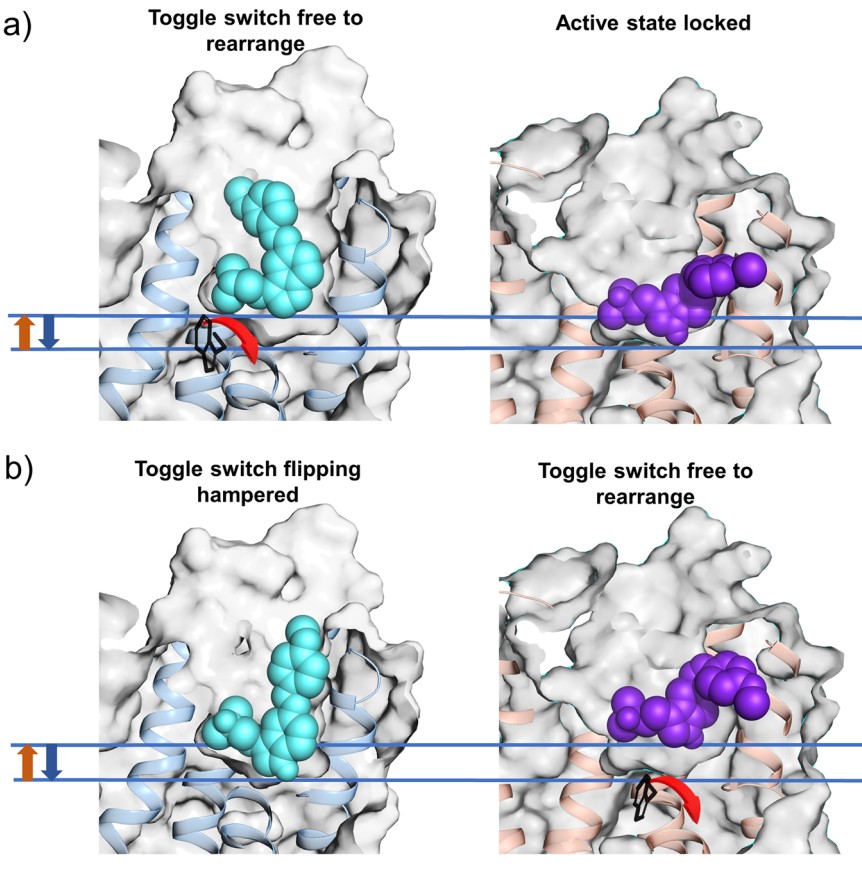

neighboring contacts rather than causing immediate ligand dissociation (Supplementary Note 4). Thus, our simulations provide a molecular rationale for the near-antagonist behavior of *trans*-**1** and thereby offer valuable insights for rational GPCR ligand design.

In this context, guiding ligands toward specific insertion depths and interaction profiles that mimic known agonists, such as by targeting or avoiding contacts with residues like Asp231 or modulating access to binding pocket extensions like the hydrophobic tunnel, may enhance potency. Such strategies could enable the development of tailored, high-efficacy 5-HT$_{2A}$ receptor agonists and precisely controlled photo-pharmacological tools. In fact, recent studies demonstrated that modifying N-benzyl-5-methoxy-tryptamine, a partial agonist of the 5−HT$_{2A}$R, by incorporating a *para*-methoxy-azobenzene moiety at the 3-position of its indole ring yielded a photoswitchable ligand with β-arrestin2-biased signaling properties[61]. The developed ligand spans through the entire orthosteric binding site; however, as observed for **1**, the *para*-methoxy substitution impedes full engagement with the hydrophobic tunnel. Nevertheless, the introduction of an N-benzyl moiety contributes additional stabilizing interactions, thereby promoting a more constrained receptor conformational ensemble. This conformational restriction is reminiscent of the uniform structural states reported for atypical chemokine receptor 3 during β-arrestin recruitment[62]. Notably, a meta-substituted analog may permit deeper insertion into the hydrophobic tunnel, further stabilizing the receptor–ligand complex; however, such enhanced binding depth may come at the expense of reduced signaling efficacy, as discussed previously. This study, combined with our findings emphasize that the ability to modulate binding depth and interaction networks through strategic ligand modifications could be key to developing functionally selective 5−HT$_{2A}$R ligands. Conclusively, this aligns with the broader goal in current 5−HT$_{2A}$ research to develop pathway-selective agonists for therapeutic benefit without inducing unwanted hallucinogenic effects[63].

## Methods
### System preparation and simulation setup
All-atom molecular dynamics simulations were performed using the human 5-HT$_{2A}$ receptor in both active and inactive conformational states. The active state model was derived from the Gq protein-bound cryo-EM structure (PDB ID: 6WHA[45]), while the inactive state used the antagonist-bound X-ray structure (PDB ID: 6A94[44]).

We retained the Gq subunit in all active-state simulations, as preliminary MDs showed that its absence caused the receptor to relax towards an inactive-like conformation. While the experimental pharmacological data from Gerwe et al.[25], were obtained using β-arrestin recruitment assays, the orthosteric binding pocket conformations are largely conserved between G protein- and arrestin-coupled states, with major differences occurring primarily at the intracellular face[64,65]. Therefore, our analysis of ligand binding poses in the Gq-stabilized active state provides relevant insights into the molecular determinants of efficacy.

Missing loops and residues in the crystal structures were modeled using MODELER 10.5[66] with templates from related GPCR structures. Protonation states of titratable residues at pH 7.0 were assigned using MCCE (Multi-Conformation Continuum Electrostatics)[67].

Each receptor system was embedded in a HEK cell-mimetic lipid bilayer to provide a physiologically relevant environment. The bilayer composition is: 26% cholesterol, 30% POPC, 31% POPE, 5% POPS, 2% POPG, and 6% SSM. Bilayers were constructed using the CHARMM-GUI membrane builder[68] with dimensions of $95 \times 95$ Å$^2$ in the x-y plane and a minimum water distance of 18 Å above and below the membrane. The systems were solvated with OPC water molecules[69] and neutralized with 0.15 M NaCl to match physiological conditions.

The AMBER ff19SB[70] and the SLipids forcefields[71] were used to treat the protein and the lipid components, respectively.

Initial binding poses of **1** and **2** were generated using Glide SP docking[72] with constraints based on the 5-HT$_{2A}$-LSD X-ray crystal structure

(PDB ID: 6WGT[45]). Key constraints included: (i) maintenance of the Asp155[3.32] salt bridge, essential for tryptamine binding; (ii) indole ring overlay with LSD (RMSD tolerance 2.0 Å) to ensure reasonable starting orientations; (iii) exclusion of poses with severe steric clashes. The top-scoring pose for each ligand-receptor combination was selected based on GlideScore and visual inspection. **1** and **2** were described with a modified version of the GAFF force field[73] validated against quantum mechanical calculations at the B3LYP-D3/6-31 G(d) level of theory (see Supplementary Note 3). Partial charges were derived with the multiconformational restrained electrostatic potential (RESP) protocol[74] as implemented in PyResp based on B3LYP/6-311 G(d,p) electrostatic potential calculations performed with Gaussian 09.

## Molecular dynamics simulation

A comprehensive set of eight different simulation systems was constructed to explore the conformational landscape of 5-HT$_{2A}$ under various conditions.

Each system underwent energy minimization using a combination of steepest descent and conjugate gradient algorithms. This was followed by a gradual heating phase from 0 to 310 K. The systems were then equilibrated in multiple stages with progressively decreasing harmonic restraints on protein backbone atoms. Production MD simulations were conducted in the NPT ensemble at 310 K and 1 atm without restraints. For each of the eight systems, three independent replicas were simulated in the microsecond timescale, yielding a cumulative simulation time of approximately 80 μs. The length of each MD is detailed in Table S1. Temperature was maintained using the Nosé-Hoover thermostat[75] with a coupling constant of 1.0 ps, and pressure was controlled using the Parrinello-Rahman barostat[76] with a coupling constant of 2.0 ps and a compressibility of $4.5 \times 10^{-5}\,\text{bar}^{-1}$. The LINCS algorithm[77] was used to constrain bonds involving hydrogen atoms, allowing for a 2 fs integration time step. Non-bonded interactions were calculated using a cutoff of 12 Å, and long-range electrostatic interactions were treated with the Particle Mesh Ewald (PME) method.

All MD simulations were performed using the GROMACS software package (version 2024.3)[78].

## Data availability

The data supporting all the findings of this study are available within the article and its supplementary files. Source data for all the analyses done on both inactive and active receptors are provided as Supplementary Data 1 and 2. The forcefield parameters for the *trans* and *cis* isomers, the first and last frames of all unbiased MD simulations (production), as well as the index, topology, and production mdp files have been deposited to the Zenodo repository[79].

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

## Acknowledgements

V.W., G.S., F.N., P.C., and G.R. gratefully acknowledge the computing time provided to them at the NHR Center NHR4CES at RWTH Aachen University (project number p0022782). This is funded by the Federal Ministry of Education and Research, and the state governments participating on the basis of the resolutions of the GWK for national high performance computing at universities. G.R. and P.C. acknowledge the Helmholtz European Partnering Project—Innovative high-performance computing approaches for molecular neuromedicine. G.R. further acknowledges the Federal Ministry of Education and Research (BMBF) and the state of North Rhine-Westphalia as part of the NHR Program. H.G. acknowledges the support by the International Doctoral Program "Receptor Dynamics: Emerging Paradigms for Novel Drugs" funded within the framework of the Elite Network of Bavaria (ENB). Financial support by the German Research Foundation (DFG) under DE1546/10-1 is gratefully acknowledged. This work was mainly supported under the Helmholtz Metadata Collaboration (Grant "MetaSupra") and by the Volkswagenstiftung (Az. 9A868). The authors thank Prof. Benedetta Mennucci for revising the manuscript.

## Author contributions

G.R. and P.C. acquired funding; G.S. and F.N. prepared the system and the parametrization of the ligand; V.W., G.S., and F.N. performed the molecular dynamics simulations and data analysis; G.R., P.C., V.W., G.S., and F.N. designed the research; M.D. and H.G. designed the experimental part; all authors contributed to the discussion; all authors wrote and edited the paper; all authors approved the final version of the paper.

## Funding

## Competing interests

The authors declare no competing interests.
