## [Transparent Peer Review file · Communications Chemistry]

Binding pose depth modulates photoswitchable ligands' efficacy at the 5-HT_{2A} receptor

Corresponding Author: Professor Paolo Carloni

Version 0:

Reviewer comments:

Reviewer #1

(Remarks to the Author)

This project investigated two azobenzene-based ligands differing by a methoxy substituent position, one is compound 1 and the other is compound 2, binding with active/inactive 5-HT_{2A} receptor using MD simulation methods. In total 80 microsecond-long simulations using Gromacs program were conducted on 8 systems, with several replicas for each system. I have the following concerns.

- 1). Since each simulation has several replicas, do those simulations lead to converged result for each system? Is there any deviation observed? What is the reason for that?
- 2). Based on the simulation method, there are constraints on the complex structure. Are these constraints used in the microsecond-long simulations? If authors want to observe the dynamics and conformational change of the receptor, or the switch from inactive state to active state, then some constraints should be removed. If those constraints were removed, what will happen?
- 3). One of the important conclusions drawn in this work is about ligand insertion depth. However, how the insertion depth of ligand into receptor was calculated? Can the authors show the insertion depth result for each simulation system quantitatively to support what they described in the manuscript?
- 3). In this work, hydrogen bond analysis was conducted. Which program was applied to do that? What is the distance threshold and angle cutoff in hydrogen bonds calculation?
- 4). In the simulation method part, more details are needed to show how the dihedral angle, RMSDi, toggle movement, and salt bridges were calculated in this work.

Reviewer #2

(Remarks to the Author)

Weber et.al studied dynamics of binding of photoswitchable ligands' efficacy at the 5-HT_{2A} receptor. They docked the ligands into the binding pocket of the receptor with different activation states and examined the dynamics using atomistic dynamics simulations of three replicates for each system studied. Some of the pair distances presented as Supp.Info highly fluctuate among the replicates of the same system thus making it difficult to infer system-specific rearrangements. From that perspective, free energy calculations can be thought of as alternatives. If not maybe key residues reported might be mutated to alanine and perform simulations to examine their impact on the system-specific conformational rearrangements.

Another point is that the authors modeled the active state using Gq protein-bound complex. It is well-known that interactions between the ligand and receptor binding residues are modulated depending on the effector. It is not clear if which effector, G protein or b-arrestin, as well as subtype of that effector is coupled to the receptor when complex with the ligands studied. Inclusion of relevant data should be informative.

Version 1:

Reviewer comments:

Reviewer #1

(Remarks to the Author)
Acceptable.

Reviewer #2

(Remarks to the Author)
The manuscript in the current version is suitable for publication.

Reviewers' comments:

Reviewer #1 (Remarks to the Author):

This project investigated two azobenzene-based ligands differing by a methoxy substituent position, one is compound 1 and the other is compound 2, binding with active/inactive 5-HT_{2A} receptor using MD simulation methods. In total 80 microsecond-long simulations using Gromacs program were conducted on 8 systems, with several replicas for each system. I have the following concerns.

1). Since each simulation has several replicas, do those simulations lead to converged result for each system? Is there any deviation observed? What is the reason for that?

Most of the properties calculated converge to very similar values across replicas (i.e., differences within the standard deviation of the average value). However, some deviations were observed and are discussed in the original manuscript and Supplementary Note 2. To further highlight these findings and explain possible reasons for them, we have now included a summary of these deviations in the main text.

The binding modes observed across replicas were highly consistent, with only replica-specific deviations representing local rearrangements rather than substantive divergence in conclusions.

Added to main manuscript:

"While the binding modes described above represent the dominant conformations observed across replicas, our simulations also revealed replica-specific secondary poses detailed in Supplementary Note 2. For trans-1 in the inactive receptor, transient disruption of the Asp231^{5.35} interaction allowed the ligand to tilt toward TM5, forming brief contacts with Ser239^{5.44} and Gly238^{5.43}. One replica of trans-2 sampled a vestibular configuration characterized by an upward shift within the binding pocket and weakening of the Asp155^{3.32} salt bridge. For cis-1, rotation around the diazo bond in one replica disrupted the Ser242^{5.46} and Asn343^{6.55} hydrogen bonds, inducing a reorientation of the indole ring toward Ser239^{5.44} or Gly238^{5.43}. cis-2 showed the largest conformational variability: in two replicas, the ligand transiently disengaged from the conserved Asp155^{3.32} anchor, drifting into the extracellular vestibule and adopting a weakly bound, solvent-stabilized pose. These alternative configurations reflect the inherent conformational flexibility of the receptor-ligand systems, leading to local rearrangements rather than substantive divergence between simulations."

2). Based on the simulation method, there are constraints on the complex structure. Are these constraints used in the microsecond-long simulations? If authors want to observe the dynamics and conformational change of the receptor, or the switch from inactive state to active state, then some constraints should be removed. If those constraints were removed, what will happen?

We apologize for not being clear on this point. We only used constraints in the initial heating and equilibration phase, not in the productive microsecond-long MD simulations. The initial phase (heating and equilibration) was not considered in the analyses presented in the manuscript. We have now clarified this point in the Methods section.

Added to Methods section:

"Production MD simulations were conducted in the NPT ensemble at 310 K and 1 atm without restraints."

3). One of the important conclusions drawn in this work is about ligand insertion depth. However, how the insertion depth of ligand into receptor was calculated? Can the authors show the insertion depth result for each simulation system quantitatively to support what they described in the manuscript?

The ligand insertion depth was calculated by aligning the receptor structures with the respective ligands to a reference cryo-EM active state structure in complex with LSD (PDB ID: 9AS3 for active, 6WGT for inactive-like as it exhibits many structural features of the inactive receptor structure) and by calculating the RMSD between the indole moiety of LSD and the indole moiety of the respective ligand (RMSD_i). As the indole moiety interacts with the bottom of the orthosteric binding pocket, this serves as a determinant for the ligand insertion depth relative to LSD. This procedure was already reported in the original manuscript.

Following the reviewer's suggestion, we now report a comprehensive table and figure with the insertion depth results for each simulation system.

System	Receptor State	RMSD_i ≤ 2.0 Å	RMSD_i > 2.0 Å	Interpretation
trans-1	Inactive	45%	55%	Deeper than LSD
trans-1	Active	85%	15%	LSD-like
trans-2	Inactive	78%	22%	LSD-like
trans-2	Active	88%	12%	LSD-like
cis-1	Inactive	75%	25%	LSD-like
cis-1	Active	92%	8%	LSD-like
cis-2	Inactive	75%	25%	LSD-like
cis-2	Active	24%	76%	Deeper than LSD

Figure: Comparison of ligand binding depths. 1) In the inactive receptor, *trans-1* is anchored near the bottom of the binding pocket, while in the active receptor *cis-1* is sterically hindered from reaching this depth. 2) *Cis-2* forms interactions near the base of the binding pocket in the active receptor, whereas *trans-2* is displaced upward in the inactive receptor.

4). In this work, hydrogen bond analysis was conducted. Which program was applied to do that? What is the distance threshold and angle cutoff in hydrogen bonds calculation? In the simulation method part, more details are needed to show how the dihedral angle, RMSDi, toggle movement, and salt bridges were calculated in this work.

We have now detailed the analysis methods in the Methods section.

Added to Supplementary Information (Supplementary Note 5: Trajectory Analysis):

"All analyses were performed exclusively on the productive portion of the MD trajectories. The productive trajectories were processed and analyzed using the CPPTRAJ program (version 6.29.13) from the AmberTools suite. For the hydrogen bond analyses, we used a cutoff of 3.2 Å and 125° for the distance and angle, respectively. Ligand insertion depth was quantified by aligning each ligand–receptor complex to the LSD-bound active-state cryo-EM structure (PDB 9AS3) and computing the RMSDi between the nine atoms of the ligand's indole core and the corresponding nine atoms of LSD's indole core. Because the indole moiety anchors the ligand at the base of the orthosteric pocket, this RMSDi provides a direct measure of vertical insertion depth relative to LSD. The same procedure was applied to the inactive-like receptor using its LSD-bound X-ray structure as the reference. The RMSDi was computed at every 0.1 ns according to:

$$RMSD_i = \sqrt{\frac{1}{9} \sum_{i=1}^9 |r_i^{lig} - r_i^{LSD}|^2}$$

The movement of the toggle switch is described by the two dihedral angles χ_1 (N–C α –C β –C γ) and χ_2 (C α –C β –C γ –C δ 1). These dihedrals define the orientation of the side chain relative to the backbone and the indole ring. While the rotamer population of χ_2 clusters between 90°

and 150°, a χ_1 rotamer population clustering around -80° is associated with the inactive state receptor while a rotamer population clustering around -160° is defined as a hallmark of the active state receptor.

A salt bridge is a noncovalent electrostatic interaction formed between oppositely charged functional groups. Salt bridges are identified when the distance between the charged heteroatoms falls below $\sim 4 \text{ \AA}$."

Reviewer #2 (Remarks to the Author):

Weber et.al studied dynamics of binding of photoswitchable ligands`efficacy at the 5-HT_{2A} receptor. They docked the ligands into the binding pocket of the receptor with different activation states and examined the dynamics using atomistic dynamics simulations of three replicates for each system studied. Some of the pair distances presented as Supp.Info highly fluctuate among the replicates of the same system thus making it difficult to infer system-specific rearrangements. From that perspective, free energy calculations can be thought of as alternatives. If not maybe key residues reported might be mutated to alanine and perform simulations to examine their impact on the system-specific conformational rearrangements.

We thank the Reviewer for this constructive suggestion. We acknowledge that some pair distances show variability among replicas, which reflects the inherent conformational flexibility of the receptor-ligand systems. Following the Reviewer's suggestion, we performed additional molecular dynamics simulations of key mutants:

Mutant	Receptor State	Ligand
D155A	Inactive	trans-1
D231P	Inactive, Active	trans-1
T160A	Inactive, Active	trans-1, cis-2

To validate our protocol prior to investigating these mutations, we performed control simulations using the well-characterized D155A (Asp3.32Ala) mutation. It is experimentally established (Kristiansen et al., 2000 10.1016/S0022-3565(24)39293-6) that Asp155 is the critical anchor for tryptamines, forming an essential salt bridge with the protonated amine. Within 1 μs of simulation, D155A did not lead to complete ligand dissociation. This is consistent with the expectation that unbinding events typically occur on long timescales due to relatively high free energy barriers, much larger than kT. However, the ligand underwent significant repositioning within the binding pocket, losing the canonical anchoring through its protonated tail that interacts with the backbone of Cys227. The indole NH group formed a water-mediated hydrogen bond with the Ala155 backbone.

In the T160A mutant, the impact of the loss of the Thr160^{3.37} contact depends on the receptor state. In the inactive receptor, the distance between the *trans*-1 methoxy group and Asp231^{5.35} shows a bimodal distribution: the dominant population (70%) maintains hydrogen-bonding distances below 3 Å, while a secondary population (~30%) shows distances of 7 Å, indicating the cleavage of this contact. In the active receptor, the Asp231^{5.35} interaction is more consistently preserved. By contrast, the hydrogen bond between the indole NH and Ser242^{5.46} remains stable regardless of receptor state and is maintained for both *trans*-1 and *cis*-2 isomers.

Figure: Violin plot of the OCH₃...Asp231^{5.35} distance in T160A.

Figure: Violin plot of the NH...Ser242^{5,46} distance in T160A for *trans*-1 (blue) and *cis*-2 (orange).

For D231P, 1-trans remains anchored within the binding pocket of the inactive receptor via Thr160^{3,37}. On the other hand, when bound to the active receptor, the attachment points are provided by either Ser242^{5,46} or Thr160^{3,37}, reproducing the results observed with the wild type.

Figure: Violin plot of the NH...Ser242^{5,46} distance in D231P.

Figure: Violin plot of the NH...Thr160^{3,37} distance in D231P

These results support our central finding that the hydrogen bond network involving Asp231^{5,35}, Thr160^{3,37}, and Ser242^{5,46} provides redundant anchoring points, and that the pharmacological differences between compounds 1 and 2 arise from their differential ability to engage this network rather than from a single critical interaction.

Added to Supporting Information (Supplementary Note 4: Mutation Studies):

"To further validate our findings, we performed additional MD simulations of selected receptor mutants. As a control, we simulated the D155A mutation, which experimentally abolishes tryptamine binding. Within 1 μ s, the ligand did not fully dissociate. This is consistent with the expectation that unbinding events typically occur on long timescales due to relatively high free energy barriers, much larger than kT . However, it underwent significant repositioning, consistent with the reported loss of affinity. Rather than anchoring through its protonated tail that now interacts with Cys227, the ligand shifts such that the indole NH group forms a water-mediated hydrogen bond with the Ala155 backbone.

In the T160A mutant, the impact of the loss of the Thr160^{3,37} contact depends on the receptor state. In the inactive receptor, the distance between the trans-1 methoxy group and Asp231^{5,35} shows a bimodal distribution: the dominant population (70%) maintains hydrogen-bonding distances below 3 Å, while a secondary population (~30%) shows distances of 7 Å, indicating the cleavage of this contact. In the active receptor, the Asp231^{5,35} interaction is more consistently preserved. By contrast, the hydrogen bond between the indole NH and Ser242^{5,46} remains stable regardless of receptor state and is maintained for both trans-1 and cis-2 isomers.

For D231P, 1-trans remains anchored within the binding pocket of the inactive receptor via Thr160^{3,37}. On the other hand, when bound to the active receptor, the attachment points are provided by either Ser242^{5,46} or Thr160^{3,37}, reproducing the results observed with the wild type.

These results support our central finding that the hydrogen bond network involving Asp231^{5,35}, Thr160^{3,37}, and Ser242^{5,46} provides redundant anchoring points, and that the pharmacological differences between compounds 1 and 2 arise from their differential ability to engage this network rather than from a single critical interaction.

Another point is that the authors modeled the active state using Gq protein-bound complex. It is well-known that interactions between the ligand and receptor binding residues are modulated depending on the effector. It is not clear if which effector, G protein or b-arrestin, as well as subtype of that effector is coupled to the receptor when complex with the ligands studied. Inclusion of relevant data should be informative.

We thank the reviewer for raising this important point. The experimental pharmacological data from Gerwe *et al.* (2022, 10.1002/anie.202203034) were obtained using β -arrestin recruitment assays. In our simulations, we retained the Gq subunit from the cryo-EM structure (PDB 6WHA) because preliminary simulations without coupled effectors showed

the receptor rapidly relaxing toward an inactive-like conformation, particularly along the TM3–TM6 axis. This was also reported by Viohl et al. in 2024 (10.1101/2024.07.23.604750). Therefore, the presence of an effector protein was necessary to maintain the receptor in a stable active conformation during the simulation.

We chose Gq specifically because (i) it corresponds to the primary signaling pathway of the 5-HT_{2A} receptor (Wallach et al, 2023 10.1038/s41467-023-44016-1), (ii) it represents the effector used in the cryo-EM structure determination, and (iii) the aim of our study was to characterize ligand binding poses in a stable active receptor conformation rather than to investigate effector-specific coupling mechanisms.

Importantly, the activation microswitches we analyzed (ionic lock disruption, TM6 outward movement, toggle switch rotation, and sodium pocket hydration) are conserved features of GPCR activation that precede effector coupling and are shared between G protein and β -arrestin pathways. Recent structural studies have shown that the orthosteric binding pocket conformations are largely similar between G protein- and arrestin-coupled states, with the major differences occurring at the intracellular face (Huang et al., 2020 10.1038/s41586-020-1953-1; Staus et al., 2020 10.1038/s41586-020-1954-0). Therefore, our conclusions regarding ligand binding depth and its relationship to efficacy should be applicable regardless of the downstream effector.

Added to Methods section:

"We retained the Gq subunit in all active-state simulations, as preliminary MD simulations showed that its absence caused the receptor to relax towards an inactive-like conformation. While the experimental pharmacological data from Gerwe et al. (2022, 10.1002/anie.202203034) were obtained using β -arrestin recruitment assays, the orthosteric binding pocket conformations are largely conserved between G protein- and arrestin-coupled states, with major differences occurring primarily at the intracellular face (Huang et al., 2020 10.1038/s41586-020-1953-1; Staus et al., 2020 10.1038/s41586-020-1954-0). Therefore, our analysis of ligand binding poses in the Gq-stabilized active state provides relevant insights into the molecular determinants of efficacy."